

# Global Attention of Transformer Empowers Montane Periglacial Lake Identification

Jinhao Xu[1], Min Feng[1,2], Yijie Sui[1], Yanan Su[1], Xuefei Zhang[3], Qinglin Wu[1,2], Zhimin Hu[1,4], Ruilin Wang[1,5]

[1]National Tibetan Plateau Data Center, State Key Laboratory of Tibetan Plateau Earth System, Environment and Resources, Institute of Tibetan Plateau Research, Chinese Academy of Sciences, Beijing 100101, China

[2]University of Chinese Academy of Sciences, Beijing 100049, China

[3]Land Satellite Remote Sensing Application Center, Ministry of Natural Resources, Beijing 100048,

China

[4]College of Geography and Tourism, Chongqing Normal University, Chongqing 401331, China

[5]College of Earth and Environmental Sciences, Lanzhou University, Lanzhou 730000, China

*Correspondence to*: Min Feng (mfeng@itpcas.ac.cn)

**Abstract.**  Montane periglacial lakes, as sensitive indicators of cryospheric change, are undergoing rapid

expansion under global warming. Investitating their evolving distribution is essential for monitoring climate understanding impacts and assessing associated geohazards. The complex topography and heterogeneous landscapes in high-mountain regions pose significant challenges for conventional methods, leading to the underdetection of small lakes, elevated false positive rates, and limited ability to discriminate between lake formation types. This study introduces a Vision Transformer (ViT)-based

framework for montane periglacial lake identification, employing a two-step process of lake boundary segmentation and type classification. By leveraging ViT's global attention mechanism, the framework captures long-range spatial and spectral relationships, enhancing contextual understanding of lakes and their surroundings. Compared to CNN-based models, the ViT-based approach achieved a mean intersection over union (MIoU) of 91.01% for segmentation and an F1-score of 89.75% for classification.



It significantly improved detection of small lakes (as small as 0.0001 km²), reduced artifacts from shadows, snow, ice, and river fragments, and provided more accurate lake type classification. Applied to the Southeastern Tibetan Plateau Gorge Region, a region with high glacial lake density and outburst flood risks, the framework identified 3,266 lakes (1,708 glacial and 1,558 non-glacial), surpassing existing inventories in completeness and accuracy.

**1. Introduction**

The montane periglacial environment refers to a montane surface environment dominated by cold conditions, where freeze-thaw cycles, snowmelt, and low-temperature physical weathering processes prevail (Péwé, 1969; French, 2017). Lakes formed within this environment, termed montane periglacial lakes, serve as critical indicators of cryospheric changes (Haeberli et al., 2001; García-Rodríguez et al.,

2021). Against the backdrop of global warming, the persistent net loss of ice in the cryosphere is driving the rapid expansion of montane periglacial lakes (Zhang et al., 2023; Wang et al., 2025). Montane periglacial lakes include glacial lakes directly tied to glacier retreat, making them highly responsive to climate change, with fragile moraine-dammed structures that elevate their outburst flood risk and attract widespread concern (Basnett et al., 2013; Veh et al., 2022). In contrast, non-glacial lakes, typically

formed by thermodynamic processes or precipitation, possess more stable configurations and exhibit reduced sensitivity to abrupt shifts (Luo et al., 2018; Larsen et al., 2024). Investigating the distribution, type, and evolution of montane periglacial lakes is critical for understanding cryospheric responses to climate change, managing water resources, and assessing geohazard risks.

Traditional field surveys, constrained by the inaccessibility of high-altitude environments and high

observation costs, struggle to achieve large-scale, continuous monitoring (Nagendra and Rocchini, 2008; Avtar et al., 2020). Currently, automated identification techniques based on remote sensing data have become a fundamental method for investigating montane periglacial lakes (Liaudat et al., 2012; Romashova and Chernov, 2023). However, accurately identifying montane periglacial lakes poses three major scientific challenges: (1) Small lakes dominate in number but are difficult to detect. Globally, it is

estimated that glacial lakes smaller than 0.1 km² account for over 75% of the total number (Zhang et al.,



2024b). These lakes exhibit limited information in imagery and are susceptible to sub-pixel spectral

mixing effects (Li and Sheng, 2012). Existing remote sensing studies typically set area thresholds at ≥

0.001 km², leaving smaller-scale periglacial lake populations without systematic observational data (Nie

et al., 2017; Chen et al., 2021); (2) Complex mountain topography and variable meteorological conditions

amplify remote sensing interpretation errors. Shadows cast by steep terrain exhibit low reflectance in the

visible to near-infrared bands, resembling water bodies, while seasonal snow and thin ice cover further

distort the spectral characteristics of water, adding complexity to the identification of water bodies

(Barbieux et al., 2018; Zhao et al., 2025); (3) Effective differentiation between glacial and non-glacial

lakes remains elusive. These two lake types differ significantly in formation mechanisms and disaster

susceptibility: glacial lakes depend on glacier ablation dynamics and carry high outburst risks, whereas

non-glacial lakes, governed by thermodynamics or precipitation, are structurally more stable (Huggel et

al., 2002; Buckel et al., 2018). Misclassification of these lake types may introduce systematic biases in

analysis and assessment.

However, existing identification methods exhibit systematic limitations in addressing these challenges.

Spectral thresholding, while efficient in delineating water bodies, is highly sensitive to topographic

shadows, snow, and ice cover, frequently yielding false positives or missing small lakes due to complex

illumination conditions in mountainous terrain (Zhao et al., 2018; Wang et al., 2020; Peppa et al., 2020).

Machine learning methods enhance environmental adaptability through adaptive feature selection, yet

their pixel-based frameworks struggle to resolve sub-pixel spectral mixing in small lakes and lack the

capacity to model spatial semantic relationships (Jain et al., 2015; Dirscherl et al., 2020; Nazakat et al.,

2021). Convolutional neural networks (CNNs), currently the most widely applied and effective method

for identifying montane periglacial lakes, integrate spectral and spatial features but are constrained by

the strong locality assumption of convolutional kernels, limiting their ability to capture global

relationships between key glacial lake indicators and topographic factors (Thati and Ari, 2022; Tang et

al., 2024; Sharma and Prakash, 2025). Meanwhile, automated classification of glacial versus non-glacial

lakes typically relies on proximity to glaciers (with a common threshold of 10 km), a method that

overlooks lake-specific environmental traits and hydrological connectivity, resulting in substantial errors



(Wang et al., 2013; Zhang et al., 2015). Shape-based methods incorporate additional morphological parameters but struggle with irregularly shaped mountain lakes, making accurate classification

challenging   (Feyisa et al., 2014; Jiao et al., 2012; Khandelwal et al., 2017). Spectral-based methods also face difficulties due to spectral variability caused by seasonal ice melting and the overlap of spectral signatures between ice-covered lakes and non-ice-covered lakes in specific bands, which hinders robust differentiation of their fundamental differences (Brinthan et al., 2023).

In recent years, Vision Transformer (ViT)-based methods have emerged as a promising alternative for

remote sensing image analysis, dividing imagery into fixed-size patches and leveraging a Transformer architecture to capture global dependencies. (Dosovitskiy et al., 2021). This architecture offers potential for integrating multi-band and multi-temporal data within a unified embedding space via self-attention mechanisms, which could enhance the discrimination of subtle spectral and spatial patterns among land features (Roy et al., 2023; Heidarianbaei et al., 2024). Recent applications in other geoscience domains

underscore its adaptability and relevance to montane periglacial lake identification. For instance, Peng et al. (2023) applied a Transformer-based U-Net with a Local-Global Transformer encoder to glacier extraction in the Qilian Mountains, integrating Sentinel-1 SAR, Sentinel-2 multispectral data, and DEMs, achieving an overall accuracy of 0.972 by leveraging multi-source data synergy. Similarly, Zhu et al. (2023) utilized a Swin-Transformer-enhanced DeepLabv3+ for glacier and ice shelf front detection from

SAR imagery, capturing dynamic calving events with a Mean Intersection over Union (MIoU) of 0.94, demonstrating ViT's strength in modeling long-range contextual dependencies. Nadachowski et al. (2024) employed ViT architecture for glacial landform classification using DEMs across diverse terrains, attaining up to 97.5% accuracy in distinguishing subtle morphological features. Additionally, Yan et al. (2023) developed a Transformer-based network to extract lakes from Sentinel-2 imagery in the Tibetan

Plateau, reducing cloud shadow interference with an overall accuracy of 0.9954, highlighting ViT's capacity to mitigate spectral confusion. Hou et al. (2024) introduced Hydroformer for lake level reconstruction, using frequency-enhanced attention to capture temporal dependencies with an $R^2$ of 0.813 across varied lake sizes, while Chen et al. (2024) proposed LEFormer, a hybrid CNN-Transformer model, achieving a MIoU of 97.42% for lake extraction by fusing local and global features. These studies



collectively illustrate ViT's ability to integrate multimodal data, resolve fine-scale features, and adapt to complex environments, suggesting its potential to address the challenges of detecting and classifying montane periglacial lakes and warranting further investigation in this domain.

This study proposes an intelligent identification framework for montane periglacial lakes based on ViT-based models and multi-source remote sensing data. It aims to systematically evaluate ViT's feature
representation advantages over CNNs in complex environments and elucidate the underlying physical mechanisms. These findings are expected to offer new methodological insights for the precise interpretation of diverse lake types and broader periglacial landforms. The framework will be applied in the Central-Eastern Himalaya (CEH) and the Southeastern Tibetan Plateau Gorge (STPG) region for model training and testing, respectively, to validate its effectiveness and generalization capability.
Additionally, it will facilitate a more comprehensive survey of montane periglacial lakes in the STPG region.

## 2. Materials and Methods

### 2.1 Overview

The framework proposed in this study for identifying montane periglacial lakes, as depicted in Figure 1,
consists of four key steps: data preprocessing, model training, prediction and validation, and postprocessing. At its core, this framework employs a two-stage strategy—segmentation followed by classification—to detect montane periglacial lakes, diverging from traditional semantic segmentation that simultaneously conducts segmentation and classification. This shift is driven by the challenges posed by incomplete lake representations in imagery due to cropping, coupled with the high similarity among lake
bodies and the often fragmented nature of environmental features. Such conditions can compromise classification accuracy in conventional workflows, potentially resulting in different regions of the same lake being assigned distinct types. To circumvent these issues, the proposed framework first segments lake outlines, then extends a defined area around these contours for secondary image cropping, before performing type classification. This ensures that the classification imagery encompasses both the
complete lake body and its environmental context, thereby improving classification accuracy and



consistency. Experiments were conducted using Python 3.11 and PyTorch 2.1.2 (Paszke et al., 2019) on

an NVIDIA 4060TI GPU (16 GB RAM, CUDA 12.3, cuDNN 8.9.7) and an AMD Ryzen 5 7500F CPU

(6 cores, 12 threads, 3.7 GHz).

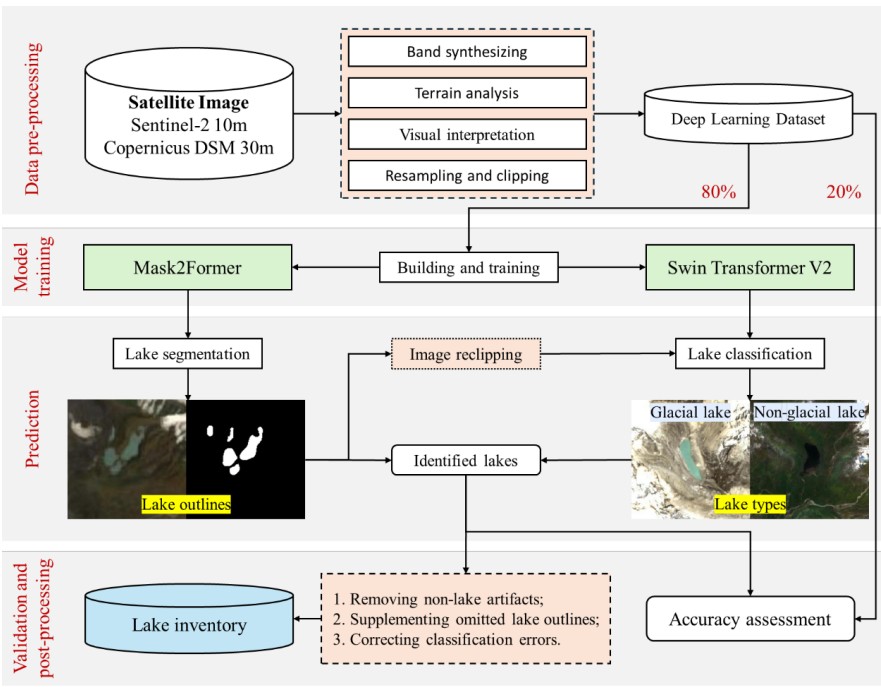

**Figure 1: Flowchart of the proposed framework.**

## 2.2 Study Sites

This study targets the CEH (Figure 2a) and the STPG region (Figure 2b), both situated along the southern

margin of the Tibetan Plateau. These regions host a high density of montane periglacial lakes and rank

among the most dynamic zones of glacial lake evolution globally (Bajracharya et al., 2007; Ahmed et al.,

2021; Furian et al., 2022). With elevations typically exceeding 4000 m asl and extensive glacier coverage,

they experience concentrated summer precipitation driven by the South Asian monsoon (Wang and

French, 1995; Zheng et al., 2000). Amid global warming and glacier retreat, rapid glacial lake expansion

has heightened the risk of glacier lake outburst floods (GLOFs) (Bajracharya and Mool, 2009; Ahmed et

al., 2021). The CEH, centered on the Himalayan main ridge, features a stepped topographic gradient,



with annual precipitation of 1,500–2,500 mm (June–September) and dense populations, posing risks to downstream communities from lake outbursts (Karki et al., 2017; Xiang et al., 2024). In contrast, the STPG region lies at the tectonic junction of the Himalayas, Hengduan Mountains, and Nyainqentanglha Range, characterized by intense tectonic activity, steep, fragmented terrain, and deep V-shaped valleys (Wang et al., 2014; Yu et al., 2020). It receives 2,500–4,000 mm of annual rainfall (May–October),

influenced by Indian Ocean moisture and the Yarlung Zangbo vapor channel, with sparse human activity yet heightened flood potential due to extreme topographic relief (Sun and Su, 2020; Chen et al., 2024b). Deep learning samples from the CEH will be used to train models, leveraging the region's moderate topographic variability and diverse lake characteristics to ensure comprehensive feature learning. The STPG region, with its extreme terrain, higher precipitation, and complex environmental conditions, will

serve as the test region to evaluate the model's performance and generalization capability. This selection enables models to address varied topographic and climatic challenges, ensuring applicability across diverse periglacial landscapes.



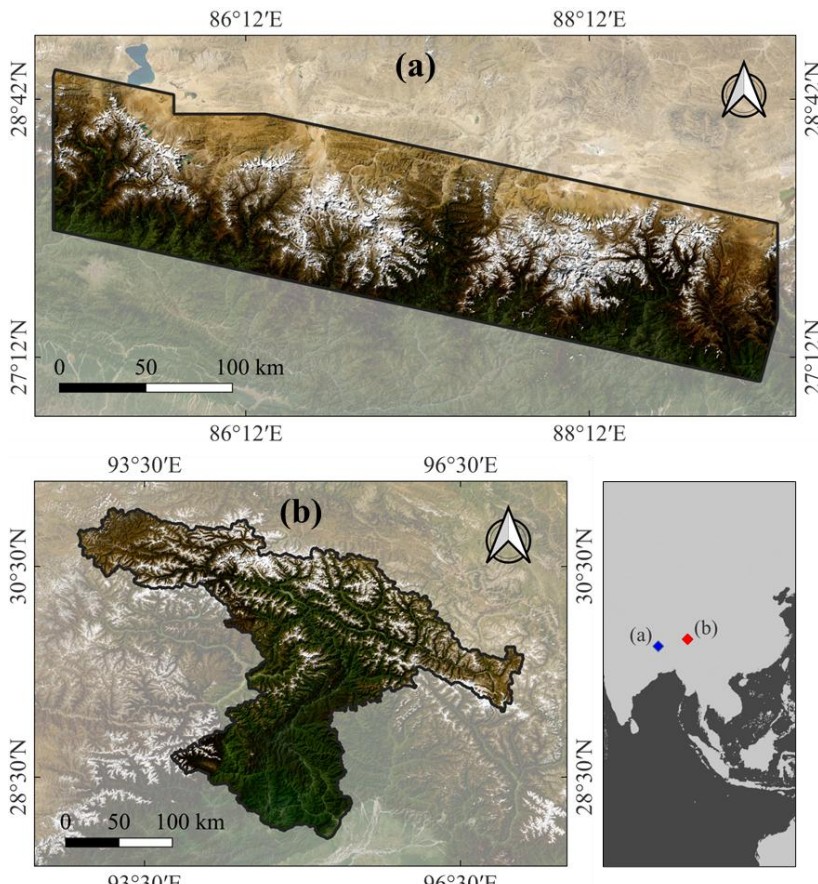

**Figure 2: The location of the study sites: (a) The CEH; (b) The STPG region. Base map sourced**

160                                       **from ESRI ArcGIS World Imagery.**

### 2.3 Data Sources and Data Pre-processing

This study utilizes 10m resolution Sentinel-2 Level-2A imagery and 30m resolution Copernicus Digital

Surface Model (DSM) as data sources. The Sentinel-2 imagery was accessed via Google Earth Engine

(https://earthengine.google.com/, last accessed: 13 April 2025), while the Copernicus DSM was obtained

from OpenTopography (https://portal.opentopography.org/, last accessed: 13 April 2025).

Sentinel-2 images (June–October 2020) were selected and processed into cloud-free composites on

Google Earth Engine to minimize snow cover and capture peak lake extent driven by monsoon and glacial



melt. The utilized bands—B2 (Blue), B3 (Green), B4 (Red), B8 (Near-Infrared), and B11 (Shortwave

Infrared)—generate RGB imagery, Normalized Difference Water Index (NDWI) (McFeeters, 1996),

Normalized Difference Vegetation Index (NDVI) (Rouse et al., 1974), and Normalized Difference Snow

Index (NDSI) (Hall et al., 1995). RGB imagery provides information of lake color, shape, and location

cues; NDWI enhances water body contrast for lake differentiation; NDVI reflects vegetation to prevent

misidentification; and NDSI highlights snow and ice to avoid confusion while informing the glacial

context. Slope and Topographic Wetness Index (TWI) (Beven and Kirkby, 1979) were derived from the

DSM using the Geospatial Data Abstraction Library (GDAL, https://github.com/OSGeo/gdal, last

accessed: 13 April 2025) and GRASS GIS (Version 8.4.1). Slope reflects the flatness of lake areas (near-

zero for lakes), while TWI indicates potential wet areas and hydrological flow paths, aiding in assessing

glacier-related lake replenishment. All data were resampled to 5-m resolution using GDAL tools and the

Lanczos resample method (Lanczos, 1950), then reprojected to EPSG:3857 (WGS 84 / Pseudo-Mercator).

Training labels were generated via visual interpretation in the CEH using RGB imagery supplemented

by NDWI, adhering to the glacial lake classification system of Yao et al. (2018). Labels comprise lake

outlines (0 for background, 1 for lakes) and types (0 for glacial lakes, 1 for non-glacial lakes), interpreted

by two researchers experienced in glacial lake studies, yielding 5,693 labels (3,995 glacial lakes, 1,698

non-glacial lakes). Data were standardized and processed into two distinct sample types for different

tasks. Segmentation samples, used for lake outline detection, incorporated RGB, NDWI, NDVI, NDSI,

and slope data. These were systematically cropped into 256×256 pixel tiles in a regular grid pattern to

ensure comprehensive coverage, yielding 4,056 positive samples (containing lakes) and 6,045 negative

samples (lacking lakes, randomly selected) to maintain data representativeness. Classification samples,

designed to distinguish glacial from non-glacial lakes, included RGB, TWI, and lake outlines. For each

lake outline, the boundary was extended outward by 1 km to form a region encompassing the lake and

its surrounding environmental context. These regions were then cropped and resized to 256×256 pixels,

balancing contextual inclusion with detail retention. The classification dataset comprised 3,995 positive

samples (glacial lakes) and 1,698 negative samples (non-glacial lakes), with sample counts aligned with

their respective labels.



### 2.4 Model Architectures and Training Parameters


Lake outline segmentation utilizes Mask2Former (Cheng et al., 2022), an advanced ViT-based model tailored for semantic segmentation, with its architecture illustrated in Figure 3a. Mask2Former enhances multi-scale feature extraction through optimized mask generation and feature interaction strategies. Its architecture consists of a backbone network for extracting multi-scale features, a Transformer decoder

that refines feature maps using self-attention and cross-attention mechanisms, and a mask prediction head that reformulates segmentation as a mask classification task. This design supports robust processing of high-resolution imagery and enables effective capture of detailed spatial patterns across diverse conditions. Lake type classification employs Swin Transformer v2 (Liu et al., 2022), an advanced hierarchical ViT-based model optimized for image classification, with its architecture illustrated in

Figure 3b. Swin Transformer v2 improves feature representation with long-spaced continuous position bias and efficient computational operations. Its architecture includes a hierarchical backbone for generating multi-scale feature maps, a shifted-window self-attention mechanism for integrating local and global contextual information, and a classification head for streamlined label prediction. This structure facilitates the efficient analysis of high-resolution remote sensing imagery, with the potential to discern

intricate spatial and contextual relationships.

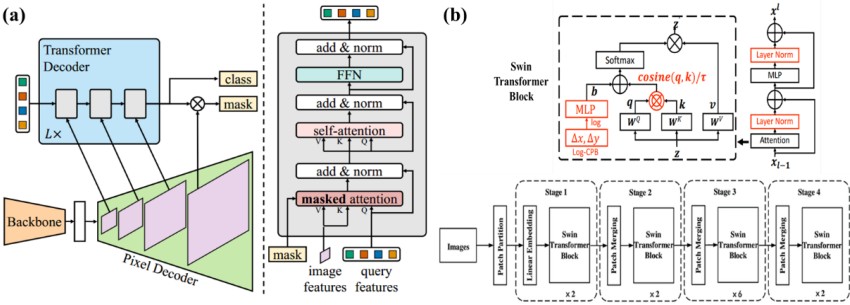

**Figure 3: (a) Architecture of the Mask2Former model (Cheng et al., 2022); (b) Architecture of the Swin Transformer v2 (Liu et al., 2022).**

Mask2Former and Swin Transformer v2 were initialized with pre-trained parameters from Cityscapes

and ImageNet-1k datasets, respectively, sourced from Hugging Face (https://huggingface.co, last accessed: 13 April 2025). Training involved 100 epochs, an 8:2 training-validation split, a batch size of



16, CrossEntropyLoss (Bridle, 1989), the AdamW optimizer (Loshchilov and Hutter, 2019), and a cosine

schedule with warmup. Samples were shuffled before each epoch to enhance generalization.

**2.5 Performance Assessment and Post-processing**

To assess the performance of the segmentation and classification models, four metrics were employed.

MIoU was utilized to evaluate the segmentation model, while precision, accuracy, and F1-score were

applied to the classification model. MIoU quantifies the overlap between the predicted and ground-truth

segmentation regions, calculated as the ratio of their intersection to their union, averaged across all

classes. This metric provides an overall measure of segmentation accuracy, with higher values indicating

superior performance. Precision represents the fraction of true positive predictions among all samples

classified as positive, reflecting the model's accuracy in identifying positive instances; a higher precision

corresponds to fewer false positives. Recall, defined as the proportion of true positives correctly

identified among all actual positive samples, measures the model's ability to detect positive instances,

with higher values indicating fewer missed positives. The F1-score, computed as the harmonic mean of

precision and recall, balances these two metrics and is particularly valuable when both accuracy and

completeness are critical, with higher values denoting a more robust and balanced model performance.

The mathematical formulations for these metrics are provided below:

$$MIoU = \frac{1}{c} \sum_{i=1}^{c} \frac{A_i \cap B_i}{A_i \cup B_i} \qquad (1)$$

$C$ represents the number of classes, $A_i$ represents the actual segmented area for the $i$th class and $B_i$

represents the predicted segmented area for the $i$th class.

$$Precision = \frac{TP}{TP + FP} \qquad (2)$$

$$Recall = \frac{TP}{TP + FN} \qquad (3)$$

$$F1\ Score = 2 \times \frac{Precision \times Recall}{Precision + Recall} \qquad (4)$$

$TP$ represents the number of samples correctly classified as positive, $FP$ represents the number of





samples incorrectly classified as positive, and *FN* represents the number of samples incorrectly classified as negative.

Following the performance assessment, the predicted lake outlines and types in the STPG region were refined through post-processing to produce the final lake inventory. Specifically, optical imagery and topographic data were integrated, with high-resolution Google Earth images used as a reference, to

remove non-lake artifacts, supplement incomplete or missing lake outlines, and correct classification errors.

### 3. Results

#### 3.1 Comparative Analysis of ViT-based and CNN-based Models for Lake Segmentation

This study trained lake outline segmentation models over 100 epochs using a sample set from the CEH

to evaluate their performance. As shown in Table 1, the Mask2Former model achieved the highest MIoU of 91.01% for lake outline segmentation. In contrast, CNN-based models, UNet and DeepLab v3+, trained with identical inputs, recorded MIoUs of 85.44% and 87.71%, respectively. Mask2Former outperformed both CNN-based models in detection rate — defined as the ratio of detected lakes to the total number of lakes — achieving 93.17% compared to 87.78% for UNet and 88.89% for DeepLab v3+.

It also generated substantially fewer non-lake artifacts (380, compared to 1,180 for UNet and 750 for DeepLab v3+).

**Table 1: Lake segmentation performance of UNet, DeepLab v3+, and Mask2Former.**

| Model | Total polygons | Detected lakes | Missed lakes | Non-lake artifacts |
|-------|----------------|----------------|--------------|--------------------|
| Mask2Former | 3610 | 3043 | 223 | 380 |
| UNet | 4674 | 2867 | 399 | 1180 |
| DeepLab V3+ | 4063 | 2903 | 363 | 750 |

To evaluate the segmentation performance across different lake sizes, detection rates of Mask2Former, UNet, and DeepLab v3+ were compared (Table 2). Results indicate that Mask2Former consistently



outperforms the other models across all size categories, with particularly pronounced advantages for

smaller lakes. For ultra-small lakes (<0.001 km²), Mask2Former achieved a detection rate of 73.42%,

notably higher than UNet (61.32%) by 19.73% and DeepLab v3+ (63.42%) by 15.77%. For small lakes

(0.001–0.01 km²), Mask2Former maintained a detection rate of 92.90%, surpassing UNet (86.07%) by

7.93% and DeepLab v3+ (87.52%) by 6.15%. For medium-sized lakes (0.01–0.1 km²), the performance

gap narrowed, with Mask2Former achieving 98.80%, outperforming UNet (96.48%) by 2.40% and

DeepLab v3+ (96.94%) by 1.92%. In large lakes (>0.1 km²), the detection rates of all three models

converged, with Mask2Former and DeepLab v3+ both reaching 99.65%, while UNet lagged slightly at

99.30%, trailing Mask2Former by 0.35%.

**Table 2: Number of lakes detected by UNet, DeepLab v3+, and Mask2Former across area ranges.**

| Area<br>Model | <0.001 km² | 0.001–0.01 km² | 0.01–0.1 km² | >0.1 km² |
|---|---|---|---|---|
| All | 380 | 1522 | 1080 | 284 |
| Mask2Former | 279 | 1414 | 1067 | 283 |
| UNet | 233 | 1310 | 1042 | 282 |
| DeepLab V3+ | 241 | 1332 | 1047 | 283 |

To evaluate segmentation performance across elevation gradients, detection rates of Mask2Former, UNet,

and DeepLab v3+ were compared (Table 3). Results show that Mask2Former consistently outperforms

the other models across all elevation ranges, with a notable advantage at lower elevations and extreme

elevations. For low-elevation lakes (<4,000 m), Mask2Former achieved a detection rate of 93.65%,

higher than UNet (88.38%) by 5.98% and DeepLab v3+ (90.56%) by 3.41%. For mid-elevation lakes

(4,000–4,500 m), Mask2Former maintained a detection rate of 92.60%, surpassing UNet (87.92%) by

5.32% and DeepLab v3+ (88.67%) by 4.43%. For high elevation lakes (4,500–5,000 m), Mask2Former's

rate reached 93.56%, outperforming UNet (89.46%) by 4.58% and DeepLab v3+ (90.15%) by 3.78%,

though the gap narrowed. For extreme elevation lakes (>5,000 m), Mask2Former sustained the highest



rate at 92.90%, exceeding UNet (85.70%) by 8.40% and DeepLab v3+ (86.87%) by 6.94%.

**Table 3: Number of lakes detected by UNet, DeepLab v3+, and Mask2Former across elevation ranges.**

| Elevation<br><br>Model | <4000 m | 4000–4500 m | 4500–5000 m | >5000 m |
|---|---|---|---|---|
| All | 551 | 662 | 1025 | 1028 |
| Mask2Former | 516 | 613 | 959 | 955 |
| UNet | 487 | 582 | 917 | 881 |
| DeepLab V3+ | 499 | 587 | 924 | 893 |

**3.2 Comparative Analysis of ViT-based and CNN-based Models for Lake Classification**

In lake type classification, Swin Transformer v2 achieved the highest F1-score of 89.75%, followed by EfficientNet (82.43%) and ResNet (82.33%) (Table 4). Swin Transformer v2 also recorded the highest

recall of 92.74% and precision of 86.94%, outperforming ResNet, which had an F1-score of 82.33%, precision of 78.46%, and recall of 88.62%, by 7.42% in F1-score, 8.48% in precision, and 4.12% in recall, indicating ResNet's greater tendency to misclassify non-glacial lakes as glacial lakes. Similarly, it surpassed EfficientNet, with an F1-score of 82.43%, precision of 80.11%, and recall of 84.89%, by 7.32% in F1-score, 6.83% in precision, and 7.85% in recall. ResNet tends to misclassify non-glacial lakes as

glacial lakes, while EfficientNet is more likely to miss glacial lakes, classifying them as non-glacial lakes.

**Table 4: Confusion matrix results for lake classification using Swin Transformer v2, ResNet, and EfficientNet.**

| Model | TP | TN | FP | FN |
|---|---|---|---|---|
| Swin Transformer v2 | 1620 | 1332 | 226 | 88 |
| ResNet | 1588 | 1261 | 297 | 120 |




| EfficientNet | 1559 | 1307 | 251 | 149 |
|---|---|---|---|---|

Classification performance was evaluated across four area ranges, with Swin Transformer v2 consistently
achieving the highest F1-scores(Table 5). For ultra-small lakes (<0.001 km²), Swin Transformer v2

recorded an F1-score of 92.24%, precision of 90.53%, and recall of 93.62%, outperforming ResNet by
7.37% in F1-score, 10.37% in precision, and 7.66% in recall, and EfficientNet by 6.47% in F1-score,
9.82% in precision, and 6.39% in recall. In small lakes (0.001–0.01 km²), Swin Transformer v2 led with
an F1-score of 89.39%, precision of 87.69%, and recall of 91.68%, exceeding ResNet by 7.42% in F1-
score, 9.07% in precision, and 5.01% in recall, and EfficientNet by 7.47% in F1-score, 7.90% in precision,

and 6.71% in recall. For medium-sized lakes (0.01–0.1 km²), Swin Transformer v2 achieved an F1-score
of 89.74%, precision of 84.31%, and recall of 93.89%, surpassing ResNet by 6.61% in F1-score, 7.40%
in precision, and 6.94% in recall, and EfficientNet by 5.98% in F1-score, 4.55% in precision, and 10.10%
in recall. For large lakes (>0.1 km²), Swin Transformer v2 posted an F1-score of 90.40%, precision of
85.61%, and recall of 94.17%, ahead of ResNet by 5.28% in F1-score, 5.14% in precision, and 8.34% in

recall, and EfficientNet by 5.17% in F1-score, 2.82% in precision, and 10.00% in recall. Swin
Transformer v2 consistently showed superior performance across all lake sizes, with its advantage most
pronounced in ultra-small and small lakes, and slightly reduced in medium and large lakes.

**Table 5: Confusion matrix results across area ranges for Swin Transformer v2, ResNet, and
EfficientNet.**

| Area | Model | TP | TN | FP | FN |
|---|---|---|---|---|---|
| | Swin Transformer v2 | 220 | 122 | 23 | 15 |
| <0.001 km² (n=380) | ResNet | 202 | 95 | 50 | 33 |
| | EfficientNet | 205 | 96 | 49 | 30 |
| 0.001–0.01 km² (n=1522) | Swin Transformer v2 | 805 | 531 | 113 | 73 |
| | ResNet | 761 | 437 | 207 | 117 |





| | | | | | |
|---|---|---|---|---|---|
| | EfficientNet | 746 | 455 | 189 | 132 |
| | Swin Transformer v2 | 446 | 522 | 83 | 29 |
| 0.01–0.1 km² (n=1080) | ResNet | 413 | 481 | 124 | 62 |
| | EfficientNet | 398 | 504 | 101 | 77 |
| | Swin Transformer v2 | 113 | 145 | 19 | 7 |
| >0.1 km² (n=284) | ResNet | 103 | 139 | 25 | 17 |
| | EfficientNet | 101 | 143 | 21 | 19 |

Classification performance was evaluated across four elevation ranges, with Swin Transformer v2 consistently achieving the highest F1-scores(Table 6). At low-elevation lakes (<4,000 m), Swin Transformer v2 achieved an F1-score of 88.2%, precision of 86.8%, and recall of 89.6%, outperforming ResNet by 13.9% in F1-score, 13.2% in precision, and 14.8% in recall, and EfficientNet by 10.8% in F1-score, 9.5% in precision, and 12.2% in recall. In mid-elevation lakes (4,000–4,500 m), Swin Transformer

v2 recorded an F1-score of 92.1%, precision of 90.5%, and recall of 90.5%, exceeding ResNet by 8.6% in F1-score, 11.8% in precision, and 5.5% in recall, and EfficientNet by 9.1% in F1-score, 11.1% in precision, and 7.1% in recall. At high-elevation lakes (4,500–5,000 m), Swin Transformer v2 maintained an F1-score of 89.7%, precision of 84.1%, and recall of 89.7%, surpassing ResNet by 5.6% in F1-score, 6.5% in precision, and 4.7% in recall, and EfficientNet by 4.8% in F1-score, 5.4% in precision, and 3.8%

in recall. At extreme-elevation lakes (>5,000 m), Swin Transformer v2 achieved an F1-score of 91.2%, precision of 90.9%, and recall of 93.2%, ahead of ResNet by 3.6% in F1-score, 6.7% in precision, and 0.9% in recall, and EfficientNet by 3.8% in F1-score, 5.3% in precision, and 2.0% in recall. Swin Transformer v2 consistently demonstrated superior performance across all elevation ranges, with its advantage most pronounced in low and mid-elevation lakes, and more comparable to ResNet and

EfficientNet in high and extreme-elevation lakes.

**Table 6: Confusion matrix results across elevation ranges for Swin Transformer v2, ResNet, and EfficientNet.**



| Elevation | Model | TP | TN | FP | FN |
|---|---|---|---|---|---|
| <4000 m (n=551) | Swin Transformer v2 | 138 | 376 | 21 | 16 |
| | ResNet | 96 | 349 | 48 | 58 |
| | EfficientNet | 84 | 362 | 35 | 70 |
| 4000–4500 m (n=662) | Swin Transformer v2 | 210 | 401 | 29 | 22 |
| | ResNet | 190 | 355 | 75 | 42 |
| | EfficientNet | 181 | 365 | 65 | 51 |
| 4500–5000 m (n=1025) | Swin Transformer v2 | 515 | 361 | 116 | 33 |
| | ResNet | 465 | 323 | 154 | 83 |
| | EfficientNet | 473 | 331 | 146 | 75 |
| >5000 m (n=1028) | Swin Transformer v2 | 721 | 182 | 72 | 53 |
| | ResNet | 728 | 125 | 129 | 46 |
| | EfficientNet | 712 | 140 | 114 | 62 |

### 3.3 Montane Periglacial Lakes in the STPG region

The proposed framework identified 3,266 montane periglacial lakes in the STPG region, comprising

1,708 glacial lakes and 1,558 non-glacial lakes (Figure 4). Their spatial distribution exhibits significant

variability. Glacial lakes are predominantly aligned with the glacier systems of the Nyainqêntanglha

Range, Himalayas, and Hengduan Mountains, extending from northwest to southeast, reflecting the

primary role of glacial activity in their formation. Non-glacial lakes, conversely, are concentrated in non-

glaciated regions, primarily in the northwest, central-north, and southern sectors of the study area,

indicating distinct geomorphological controls on their distribution.



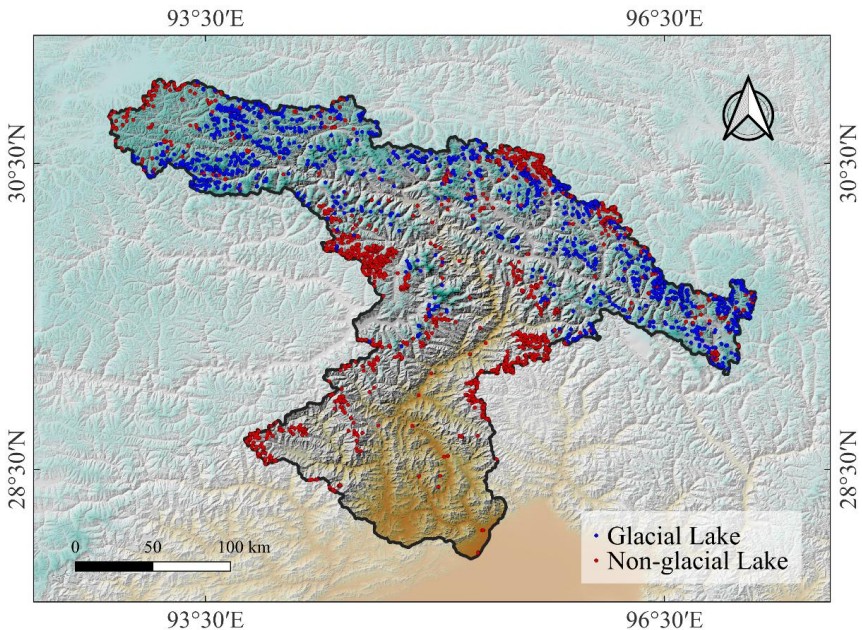

**Figure 4: Montane periglacial lakes in the STPG region (blue dots represent glacial lakes, and red dots represent non-glacial lakes).**

The identified lakes collectively occupy a total area of 175.6 km², with a mean area of 0.054 km² per lake. Glacial lakes account for 89.3 km² and non-glacial lakes for 86.3 km², with mean areas of 0.052 km² and 0.055 km², respectively, suggesting comparable scales despite differing origins. Lake size distribution is skewed toward smaller dimensions (Figure 5a): approximately 79.6% of lakes range between 0.001 and 0.1 km². Lakes <0.001 km² are relatively uncommon, while those in the 0.001–0.01 km² and 0.01–0.1 km² ranges increase markedly in frequency; lakes >0.1 km² are scarce. This pattern implies that local topography and hydrological conditions favor the formation of small to medium-sized lakes. Elevationally, the lakes occur at a mean elevation of 4,600 m, with glacial lakes averaging 4,822 m and non-glacial lakes 4,356 m, highlighting their association with high-elevation glacial environments. Lake abundance exhibits an upward trend with elevation (Figure 5b), increasing from sparse occurrences below 4,000 m to progressively higher frequencies in the 4,000–4,500 m, 4,500–5,000 m, and >5,000 m ranges.



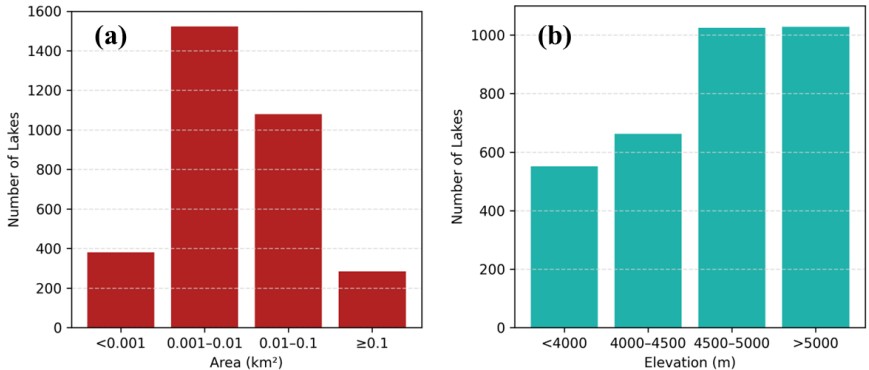

**Figure 5: (a) Size-classified and (b) elevation-classified distributions of montane periglacial lakes in the STPG region.**

## 4. Discussion

### 4.1 Performance Advantages of ViT-based Models over CNN-based models

Experimental results demonstrate that ViT-based models significantly outperform CNN-based models in the segmentation and classification of periglacial lakes in mountainous regions. The ViT-based model excels at preserving lake boundary integrity during detection, whereas CNN-based models frequently exhibit boundary loss when processing mixed pixels, particularly in shallow zones near lake margins (as shown in Figure 6a). This disparity arises from ViT's self-attention mechanism, which constructs feature representations from a global perspective, capturing long-range pixel dependencies to delineate continuous lake boundaries accurately. In contrast, CNN-based models, constrained by localized convolutional kernels, struggle to adapt to the diverse boundary morphologies prevalent in complex terrains. To enhance information complementarity, the experiment incorporated multisource remote sensing data. For instance, when lake morphology is indistinct in RGB imagery, NDWI provides clearer boundary cues; when NDWI confounds lakes with shadowed slopes, slope data facilitates differentiation. Consequently, common non-lake artifacts caused by shadows and glacial snow are substantially suppressed in the results of both ViT-based model and CNN-based models. However, in river channel scenarios, CNN-based models still face significant challenges, often misclassifying fragmented rivers as



370     lakes due to an over-reliance on local texture features (as shown in Figure 6b). In contrast, the ViT-based

        model leverages semantic scene understanding to effectively distinguish lake from non-lake features,

        markedly reducing artifact interference.

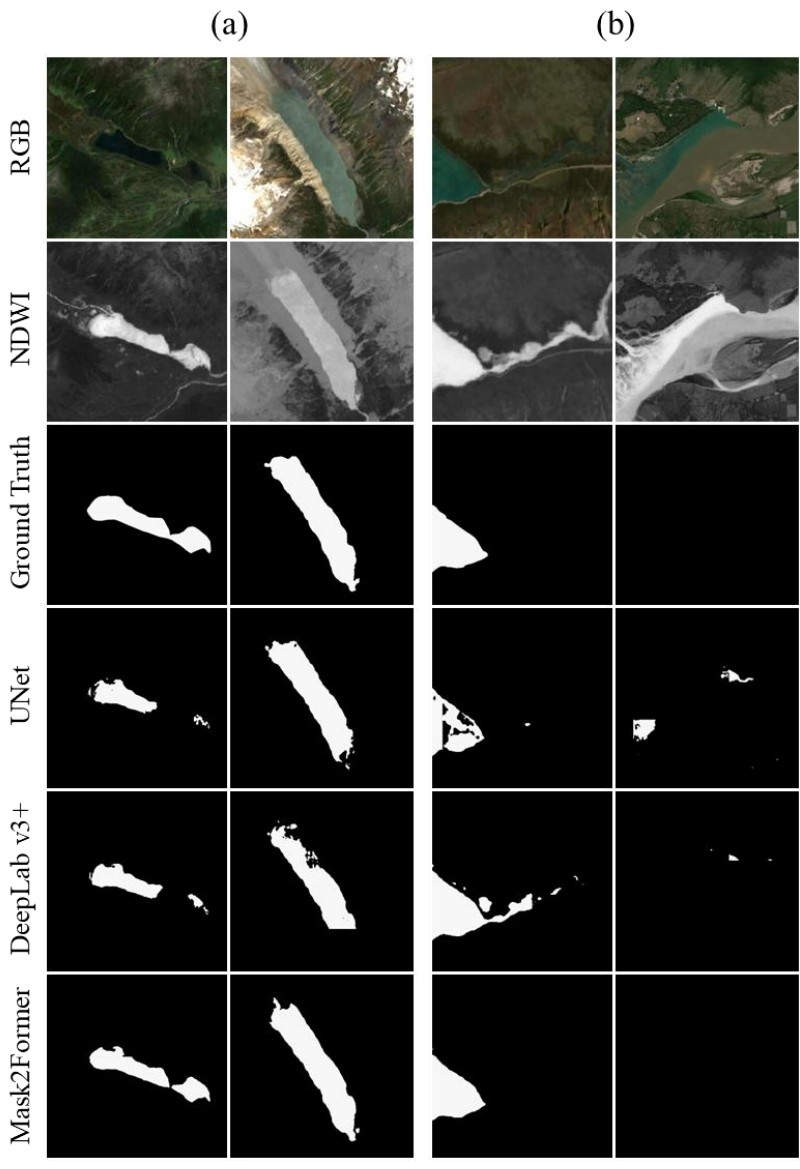



**Figure 6: Comparison of the ViT-based model and CNN-based models performance in montane** periglacial lake identification: (a) boundary detection integrity in mixed pixel scenarios and (b) artifact suppression in river channel scenarios.

In lake size categories, the ViT-based model exhibits pronounced superiority, particularly for ultra-small lakes (<0.001 km²), which often span just a few pixels in remote sensing imagery and are easily confused with surrounding vegetation or bare land. CNN-based models, constrained by fixed kernel sizes, struggle to resolve subpixel spectral mixing, resulting in misclassification or omission. The ViT-based model, through adaptive multiscale feature extraction, enhances detection rates and accuracy for these subtle targets. However, as lake size increases (>0.1 km²), the performance gap narrows, as larger spatial extents provide sufficient context for CNN-based models to achieve comparable segmentation and classification.

Across elevation ranges, the ViT-based model exhibits exceptional robustness, with particularly pronounced advantages over CNN-based models at low (<4,000 m) and extreme (>5,000 m) elevations. At lower elevations, dense vegetation cover often reduces lake visibility, while at extreme elevations, interference from snow and ice further obscures lake identification. CNN-based models, constrained by fixed local receptive fields and sensitivity to textural variability, struggle to effectively extract lake features in such complex environments, leading to frequent false positives. In contrast, the ViT-based model employs attention-driven feature selection to distinguish targets from backgrounds, enhancing segmentation and classification accuracy while improving generalization capacity. By modeling morphological continuity and spatial structure, the ViT-based model demonstrates robust lake identification under highly heterogeneous conditions, significantly reducing uncertainty.

**4.2 ViT-Enhanced Lake Inventory Completeness and Classification Accuracy**

Compared to the previous lake inventories from 2020 (Zhang et al., 2024a, b), this study demonstrates notable improvements in both the number of identified lakes and the overall completeness of the inventory. G. Zhang et al. (2024), using a combination of water body indices and visual interpretation, identified 569 glacial lakes, without including non-glacial lakes. T. Zhang et al. (2024), based on visual interpretation, documented 610 glacial lakes and 427 non-glacial lakes, totaling 1,037 lakes. In contrast,





this study mapped 1,708 glacial lakes and 1,558 non-glacial lakes, yielding a total of 3,266 lakes, approximately three times more than that of T. Zhang et al. (2024) and six times more than that of G. Zhang et al. (2024) (Table 7). These results highlight substantial advances in both the quantity and spatial completeness of lake mapping for the region. This study employs a ViT-based intelligent lake

identification framework, markedly enhancing the detection of small lakes and providing more precise boundary delineation. While the published inventories captured lakes larger than 0.001 km², the proposed framework achieves an order-of-magnitude improvement, detecting lakes as small as 0.0001 km². In contrast, traditional vision interpretation are not only time-intensive and less efficient but also prone to human-induced inconsistencies, often resulting in omitted lakes or inaccurate boundaries (Blaschke,

2010; Lillesand et al., 2015). For the inventory by G. Zhang et al. (2024), the average area of missed lakes is 0.035 km², while the average area of detected lakes is 0.129 km². Similarly, T. Zhang et al. (2024) missed lakes with an average area of 0.029 km², compared to 0.098 km² for those detected. These differences highlight the superior capability of ViT-based models for capturing smaller lakes that were previously overlooked. These small glacial lakes, which form prolifically during glacial ablation, play a

critical role in glacial lake outburst flood (GLOF) risk assessments (Yao et al., 2014; Zhang et al., 2022). Given their abundance and widespread distribution, their potential failure poses severe threats to downstream regions, emphasizing the importance of their accurate detection.

**Table 7: Comparison between this study and the published inventory.**

| Inventory | Glacial Lake | Non-glacial Lake | Sum |
|---|---|---|---|
| G. Zhang et al. (2024) | 569 | 0 | 569 |
| T. Zhang et al. (2024) | 610 | 427 | 1037 |
| This Study | 1708 | 1558 | 3266 |

Lake type classification typically relies on proximity to glaciers, an intuitive yet often flawed method

that assumes proximity is a reliable indicator of glacial origin, leading to frequent misclassifications. To evaluate its performance, this study applied the 10 km buffer method using glacier outlines from the Second Chinese Glacier Inventory v1.0 (Guo et al., 2015) to classify lakes in the STPG region, and





compared the results with ViT-derived classifications. The analysis shows that this distance-based method significantly overestimates both the number and spatial extent of glacial lakes. As shown in

Figure 7, in an area approximately 30 km east of Motuo, more than 30 non-glacial lakes were misclassified as glacial simply because they fell within the 10 km buffer, despite being stable water bodies without glacial meltwater input. This discrepancy underscores the limitations of proximity-based classification. In contrast, the ViT-based model integrates spectral, morphological, and environmental features through its self-attention mechanism, enabling more accurate differentiation

between glacial and non-glacial lakes—even in complex periglacial settings. This improvement reduces uncertainty in glacial lake inventories and enhances the reliability of climate risk assessments, providing a stronger basis for targeted disaster mitigation strategies.

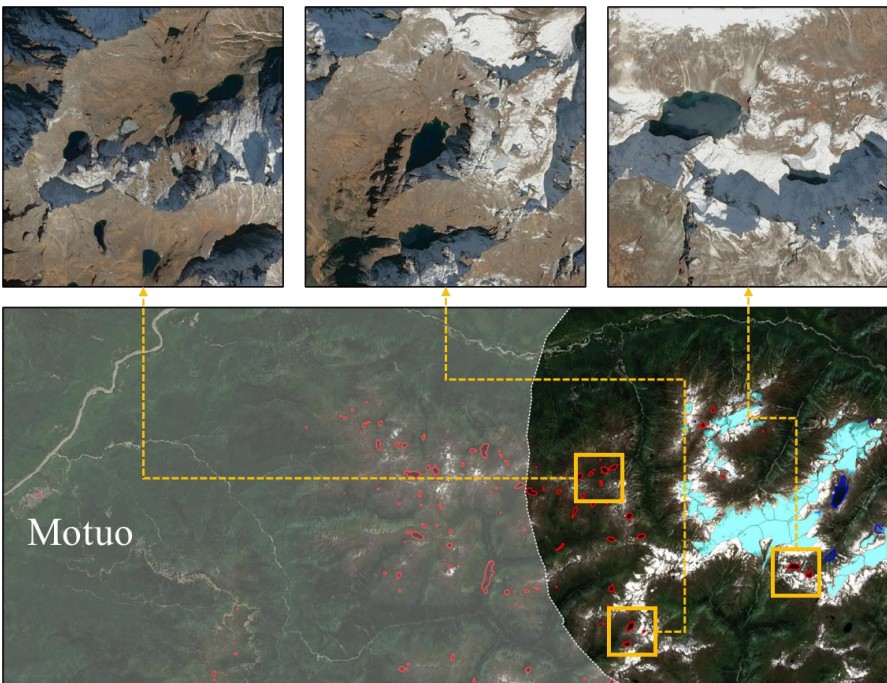

**Figure 7: Distribution of montane periglacial lakes in an area located approximately 30 km east**

**of Motuo County, Tibet, China (blue lines represent glacial lakes, red lines represent non-glacial**

**lakes, Cyan polygons represent glacier extents, and the white translucent mask delineates areas**

**located beyond 10 km from glaciers.). Base map sourced from ESRI ArcGIS World Imagery and**



**Yandex Maps.**

### 4.3 Limitations and perspectives

Despite the notable strengths of the ViT-based model in glacial lake identification, several limitations

persist in this study. Glacial lakes situated far from glaciers—up to 15 km, as noted by Yao et al. (2018)—

are occasionally misclassified due to the constrained spatial context resulting from cropped input images.

While ViT's global attention mechanism partially mitigates this issue by capturing broader dependencies

compared to conventional CNNs, the challenge of modeling long-distance spatial relationships suggests

a need for multi-scale methods, such as hierarchical Transformer architectures, to enhance accuracy.

Additionally, summit snow cover introduces spectral confusion with glaciers in low-resolution imagery,

leading to misclassifications of lakes. Incorporating time series data to account for seasonal snow

variations could refine classification by providing temporal context, potentially reducing errors by

distinguishing transient snow from permanent glacial features. Integrating multi-temporal imagery could

address this by supplying historical context, enabling more precise identification of lakes with glacial

origins.

### 5. Conclusion

This study proposed an intelligent framework for identifying montane periglacial lakes using ViT-based

models. Compared to CNN-based models, ViT-based models demonstrated superior segmentation

accuracy and classification robustness across diverse lake sizes and elevations. It effectively detected

small lakes—often missed by CNN-based models—while minimizing false positives, such as mountain

shadows and river fragments. The ViT-based model also distinguished glacial from non-glacial lakes

with greater precision than the traditional glacier-proximity-based method, which is prone to

Overestimation.

When applied to the STPG region, the framework produced an inventory of 3,266 lakes, comprising

1,708 glacial and 1,558 non-glacial lakes. This inventory exceeded the completeness of published

datasets, highlighting the efficacy of ViT-based models in complex alpine terrains. The resulting dataset



offers high-quality data to support the analysis of lake evolution and the assessment of climate-driven hydrological risks in glaciated regions.

*Author contribution.* JX planned the study and performed the experiments. JX analyzed the data and wrote the manuscript draft. JX, MF, YS, and QW reviewed and edited the manuscript. MF supervised the study, managed project administration, and secured funding. YS, XZ, ZH, and RW curated the data.

*Competing interest.* The authors declare that they have no known competing financial interests or personal relationships that could have appeared to influence the work reported in this paper.

*Financial support.* This work was supported by the National Key Research and Development Program of China: [grant number 2023YFF0725005]; the TPESER Youth Innovation Key Program: [grant number TPESER-QNCX2022ZD-04]; the National Natural Science Foundation of China: [grant number 42301160]; the Science and Technology Department of Tibet Program: [grant number XZ202301ZY0035G].

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
