# Peer review of "Global Attention of Transformer Empowers Montane Periglacial Lake Identification"

_EGUsphere, 2025_

## Author Comment (AC1)

The proposed manuscript "Global Attention of Transformer Empowers Montane Periglacial Lake Identification" seeks to advance the remote-sensing based detection and classification of montane periglacial lakes. Accurate inventories of these lakes are of particular relevance as these lakes are indicators of climate change, important sources of fresh water, and pose geohazard risks through GLOFs. The authors identify three main challenges in the remote-sensing based detection of periglacial lakes, which are the difficult detection of very small lakes, spectral confusion due to topographical shadows and similar land surface classes, and the discrimination between glacial and non-glacial lakes. To address these challenges, the authors propose a two-stage classification approach, in both of which Vision Transformer (ViT) models replace more established models.

First, lakes in a Himalayan study region are detected from a Sentinel-2 mosaic using image segmentation. For this the authors propose the ViT model Mask2Former. Second, the identified lake shapes are analysed in their original environmental context to semantically classify them as either glacial or non-glacial. For this task, the authors propose the Swin Transformer v2 model. The models are trained in one region and applied and tested in a second to avoid overfitting and ensure transferability. The model results are compared to those of different established convolutional neural networks (CNN) architectures, and the proposed framework appears to yield better results throughout. The final mapping product for the validation region is furthermore compared to two different lake mapping products. The new mapping approach detects a significantly larger amount of lakes thank the comparison datasets, which the authors attribute to the ability of their framework to detect particularly small lakes.

**General comments**

The manuscript has a clear approach, is generally well structured and concise. The methodology of comparing a newly developed framework to existing ones is suitable. The discussion does well in explaining the performance of ViT compared to CNN based on the different model architectures. The presented results are in so far relevant, as they seem to be a significant improvement in comparison to established lake mapping methods (e.g. the U-Net) in montane areas. Even though the study is driven rather by a methodological instead of a geoscientific research question, I feel that with some rework it can be a valuable contribution to the cryosphere research community as it demonstrates a way to generate comprehensive inventories of periglacial lakes.

However, I think the paper needs some major revisions before publication. A major point is that the authors need to elaborate more on their methodology to facilitate transparency of their experiments and reproducibility. More details and explanation would help the geoscientific community to better understand the selected model architectures and configurations. More specific feedback on these issues can be found in the Specific Comments below. Furthermore,

I strongly encourage the authors to share the lake labels used for training and testing in an open repository, not only for transparency but also to bolster the credibility of their results. Otherwise, it will be impossible to verify these. The same is true for the programming code of the applied models should they have been modified from their original source. Finally, I find the part of the discussion that addresses the confusion of glacial and non-glacial lakes in close proximity to glaciated areas to be insufficient. This could be improved by a similar analysis as presented in the results section. Again, more specific feedback on that can be found in the Specific comments.

**Response:** We sincerely thank you for constructive feedback and for recognizing the value of our work in improving glacial lake mapping in complex montane environments. We are encouraged by the acknowledgment of our framework's performance and its potential contribution to the cryosphere research community. Following your suggestions, we will perform a major revision of the manuscript to enhance transparency and scientific rigor.

Detailed point-by-point responses to your specific comments are provided below.

**Specific comments**

Title: I feel the word "empower" is too strong, as the proposed method rather advances/improves the already working detection of periglacial lakes.

**Response:** We agree that "empowers" might convey a stronger implication than intended. In the revised manuscript, we have replaced it with "enhancing" to more accurately reflect the incremental yet significant improvements achieved by incorporating global mechanisms within the Transformer architecture.

Furthermore, in response to the concerns raised by another reviewer regarding the potential ambiguity of "periglacial lake" and "global attention," we will comprehensively revise the title to:

"Enhancing Lake Identification in Periglacial Environments by Leveraging the Global Context of Transformers."

L52-53: What is the reasoning for this exact lake size threshold? Is it sensor resolution? Is it the low relevance of lakes of such small a size? Or from a different perspective: Why is it important to also include these small lakes and develop a method, which is able to detect these? I think it is worthwhile to address this, as the proposed methods later shows its strengths at exactly this lake size.

**Response:** The 0.001 km² threshold adopted in prior studies (e.g., Nie et al., 2017; Chen et al., 2021) primarily stems from limitations in sensor spatial resolution (e.g., 30 m Landsat pixels) and practical considerations for large-scale mapping, such as reducing false positives from shadows, snow cover, or ephemeral water bodies in complex montane terrain. Smaller lakes are often omitted to prioritize accuracy in inventories focused on water resources or outburst flood risks. However, including ultra-small lakes (<0.001 km²) is crucial because they are highly sensitive indicators of cryospheric changes in periglacial environments, responding rapidly to permafrost degradation, glacier retreat, and warming. Although individually minor, their collective dynamics contribute significantly to regional hydrology, carbon cycling, and ecosystem shifts. In the revised manuscript, we will expand the explanation in the introduction.

L68: What do you mean by "adaptive feature selection" in a Machine Learning context?

**Response:** By "adaptive feature selection," we refer to the capability of machine learning algorithms to dynamically prioritize or weight informative features during the training process, thereby improving robustness to environmental variability. This adaptability helps mitigate some challenges in heterogeneous landscapes but remains limited within pixel-based frameworks. In the revised manuscript, we will rephrase the sentence for clarity as follows:

"Machine learning methods improve environmental robustness through dynamic feature weighting and selection during model training"

L101: What is "Hydroformer"? Is it a CNN? How does its architecture compare to the other introduced methods. Briefly elaborate.

**Response:** Hydroformer (Hou et al., 2024) is a pure Transformer-based sequence modeling framework (not a CNN or CNN-Transformer hybrid). It treats satellite-derived lake area time series as sequences and introduces a frequency-enhanced self-attention mechanism to better capture seasonal and long-term temporal dependencies for lake level reconstruction. We included it here to illustrate the broader versatility of Transformer architectures in lake-related studies, particularly their strong capability in modeling long-range dependencies. In the revised manuscript, we will add a brief clarifying remark in this paragraph (approximately one sentence) to distinguish its temporal-sequence nature from the image-based ViT applications, thereby improving readability.

L101-104: To be consistent with the sources you cited before: Could you briefly add in which

spatial context (location, scale) the two studies cited here were conducted.

**Response:** We agree that adding brief spatial context for the cited studies will improve consistency with the earlier examples in this paragraph and enhance readability for readers unfamiliar with these works. Hou et al. (2024) evaluated the Hydroformer on 50 lakes distributed globally, covering a wide range of lake sizes (10.11 to 18,135 km²) and climatic conditions. Chen et al. (2024) tested LEFormer on two datasets: the Surface Water dataset (SW dataset) and the Qinghai-Tibet Plateau Lake dataset (QTPL dataset). In the revised manuscript, we will incorporate short descriptions of the spatial context/scale for these two studies directly into the text.

L107: I think what's missing here is an overview about which specific shortcomings of the ViT studies cited before the proposed approach in this paper is supposed to address. Is it just the lack of application of ViTs for the detection of periglacial lakes? I can see, that ViTs have been applied before to detect lakes (and other surface features) in different contexts, but what in the cited studies makes the authors claim that ViTs are particularly suitable for this type of setting (montane periglacial)? I very much agree that it is worthwhile to investigate the suitability of ViTs for the proposed task, but the introduction chapter could be improved by providing some stronger arguments why particularly ViTs are promising.

**Response:** We appreciate the suggestion to provide a clearer overview of how the cited ViT studies motivate our exploration and to strengthen the arguments for ViT's suitability in montane periglacial environments.

The cited ViT applications demonstrate strong performance in related geoscience tasks, such as capturing long-range contextual dependencies in glacier extraction amid rugged terrain (Peng et al., 2023; Zhu et al., 2023), distinguishing subtle morphological features in diverse landscapes (Nadachowski et al., 2024), and mitigating spectral confusion in high-altitude lake detection (Yan et al., 2023; Chen et al., 2024). However, these have not yet been extended to montane periglacial lake identification, where unique challenges arise: ultra-small lake sizes, severe spectral mixing from terrain shadows, snow/ice cover, and freeze-thaw dynamics, plus the need for precise proglacial lake classification in heterogeneous settings. ViT's global attention mechanisms are particularly promising here, as evidenced by their success in modeling extensive spatial relationships and integrating multi-spectral/multi-temporal data in analogous complex environments (Hou et al., 2024 for temporal dependencies).

In the revised manuscript, we will add a concise paragraph to emphasize these environment-specific challenges and ViT's potential, based on the cited evidence, to explore its applicability.

L110: The claim that the study "elucidates the underlying physical mechanisms" (of what?) is too bold. This is not at all addressed in the study.

**Response:** We agree that the claim to "elucidate the underlying physical mechanisms" is overstated, as the study focuses on methodological improvements rather than direct analysis of physical processes.

In the revised manuscript, we will revise the sentence to:

"These findings are expected to offer new methodological insights for the precise identification and classification of diverse lake types, thereby facilitating improved interpretation of periglacial landforms and cryospheric changes."

Figure 1: The figure indicates an accuracy assessment on the test data of the deep learning dataset. However, there is no arrow connecting back from the accuracy assessment to the two models. Were these models tuned and optimized or just used "out-of-the-box"? This should be also addressed in the text.

**Response:** The workflow is linear by design, reflecting our actual process: both Mask2Former and Swin Transformer V2 were fine-tuned on the 80% training split (as indicated in the "Model training" section), but the held-out 20% test set was used solely for final performance evaluation and accuracy assessment. No iterative hyperparameter tuning, retraining, or optimization based on test results was performed, which is why no feedback loop is shown. This approach follows common practice in remote sensing studies with limited labeled data to ensure reproducibility and prevent overfitting.

In the revised manuscript, we will add a brief clarification to the Figure 1 caption and methods section (e.g., "The accuracy assessment was performed on the independent test set without iterative model optimization or feedback loops").

Figure 2: The third panel of the map (the overview) would be much more insightful if it provided a shaded relief of the topography. This way, readers not familiar with the region would be better able to understand the setting of the two study regions within the larger topographical context. Consider also to zoom-in a little bit (not too far) to the Himalayas and surrounding mountain ranges themselves. Too much space in this panel is wasted on regions which are not important to this study (Siberia, Australia, Indonesia etc.)

**Response:** We agree and will revise the overview panel in the revised manuscript by adding

shaded relief topography and zooming in to focus on the Himalayas and surrounding mountain ranges.

L166: You only use imagery from a single season. As a training dataset should be diverse to reflect a wide range of environmental conditions you should provide a good explanation why you focus on this limited time frame.

**Response:** Our dataset is limited to composite imagery from June–October 2020 primarily because montane periglacial lakes in the study regions are most visible and distinguishable during the ablation/summer–early autumn season, when lakes are typically ice-free, water extents are maximal, and spectral contrast with surroundings is strongest. Outside this period, persistent snow/ice cover and frozen lake surfaces often make reliable identification and labeling challenging, increasing ambiguity in ground truth annotation.

In the revised manuscript, we will expand the explanation in the data section to include this rationale and acknowledge the limitation of single-season data, noting that future work could incorporate multi-seasonal composites as more annotated data become available.

L166: During compositing, how do you account for intra-annual variability of the environment and particularly lake areas? You say you favor snow-free conditions with maximum lake extent (which is totally reasonable) but how do you control that this is reflected in the composite?

**Response:** We constructed the composites from Sentinel-2 images acquired between June and October 2020. The median composite naturally favors persistent water bodies and suppresses transient features (e.g., clouds, temporary snow), while tending to reflect near-maximum lake extents in this ablation season, as water pixels are consistently present across multiple acquisitions during peak melt. However, we acknowledge that median compositing does not strictly guarantee maximum extent for every individual lake, as local variability (e.g., due to short-term precipitation) may influence the result.

In the revised manuscript, we will add a brief explanation in the data section to clarify this process and note this inherent characteristic of median composites for seasonal environments.

L178: What is the point of upsampling 10m/30m resolution input data to 5 m? Without any additional very high-resolution data there is no information gain. Why not just stick with 10m? In fact, because the input imagery into the ViTs is tiled into tiles with a fixed number of pixels (256x256), you might be losing a lot of spatial context with the higher resolution, don't

you?

**Response:** We acknowledge that upsampling to 5 m does not introduce new information. The primary rationale for this step is boundary regularization. By increasing the spatial sampling rate, the subsequent segmentation head can perform finer spatial interpolation, which is crucial for achieving smoother and more accurate boundaries for ultra-small lakes that are highly pixelated at 10 m resolution. Regarding the concern of context loss, we emphasize that the tiled area ($256^2 \times 5^2 / 10^6 = 1.6384$ km$^2$) is large enough to contain substantial regional context, especially considering that over 90% of the lakes in our study area are smaller than 0.1 km$^2$. We will address this point in the methodology section.

L180ff: Training labels: Generating training labels is always a crucial process in ML/DL approaches. If two different experts were responsible for creating these labels, could you elaborate on any measures taken to ensure consistency between the labels? Also, I feel it would be a huge benefit to the community to make the training and validation labels available to the open public.

**Response:** The training and validation labels were primarily generated through detailed visual interpretation by the first author. To ensure consistency and high accuracy, a second researcher subsequently performed an independent quality check of the interpreted boundaries. This check involved re-examining the high-resolution imagery and cross-referencing the labels against existing lake inventories. Furthermore, we fully agree with the value of data openness; we are pleased to confirm that the training and validation label datasets will be made publicly available upon the manuscript's acceptance and publication.

L184: How were the data standardized? Which method did you use?

**Response:** We appreciate the request for clarification. All input data bands were standardized using Z-score standardization. This method scales the data to have a mean of 0 and a standard deviation of 1, which significantly aids the stability and convergence of deep learning models. We will update the manuscript to include this explicit explanation.

L195ff: As this part is very technical ML/DL language, I would recommend some reworks to cater to the geoscientific community of this journal. Specifically, I'd like to see some elaboration on how the different components/features (e.g. multi-feature extraction, self/cross attention) of the two architectures are beneficial to the tasks of segmentation and classification of periglacial lakes in a montane setting. For example, which of the challenges

described in the introduction section are addressed by choosing these model architectures and configurations.

**Response:** We completely agree that the architecture explanation must be better tailored to the geoscientific community. We will significantly revise and expand this section in the final manuscript to clearly link the ML/DL components to the challenges of periglacial lake mapping in montane settings. Specifically, we will elaborate on how: Multi-Feature Extraction addresses spectral confusion with complex terrain elements (e.g., shadows); Cross-Attention enhances feature fusion for accurate assessment of the topographic and environmental context needed for reliable glacial vs. non-glacial classification. These detailed explanations will be fully incorporated into the Methods section of the final revision.

Methods-Section: The methods section misses an entire sub-section on the additional models used for model comparison, i.e. U-Net, DeepLab V3+, ResNet, and EfficientNet. Although this section does not need to be as detailed as the (revised) section 2.4, some basic information is indeed required, such as reasoning for the choice of the comparative models, proper citation of the sources of the models, configuration of the input data for these models, and essential model hyperparameters. The reader must be able to reproduce the experiments the authors performed.

**Response:** We will add a new subsection to the Methods to clearly detail the choice rationale, citations, input configuration, and essential hyperparameters for these models as requested.

L216ff: What is the reasoning behind choosing these specific hyperparameter settings? Is there a loss curve that warrants that a training of 100 Epochs is enough?

**Response:** We appreciate the request for justification. Our hyperparameter settings are based on established best practices for Vision Transformer architectures, computational efficiency, and preliminary convergence testing.

Epochs (100): We will include the loss curves to demonstrate convergence. Preliminary tests showed model performance stabilized within 80 epochs, making 100 epochs sufficient for convergence while optimizing computational cost.

AdamW & Cosine Schedule: AdamW was chosen for its effective weight decay decoupling, crucial for training large Transformer models. The Cosine schedule with Warmup is the standard recommendation for stable training of pre-trained Transformer architectures.

Batch Size (16): This setting balances GPU memory limits and optimization stability.

We will include these detailed rationales in the revised manuscript.

Section 3.1: It is very good that the authors analyse and compare the performance of the different models for lake polygons, lake size, and elevation range using the MIoU. However, this could be complemented by an analysis of lake area, i.e. the ability by the different models to map the lake area as "completely as possible". The analysis shown in Fig 6a already goes into this direction, where you can see that although, for example, DeepLab detects a lake as an entity, it fails to completely map the lake boundary as determined by the ground-truth data. I recommend a MioU analysis based on the total number of lake pixels detected by the different approaches.

**Response:** We thank the reviewer for this valuable suggestion to assess the models' ability to "map the lake area as completely as possible." We would like to clarify that the MIoU (Mean Intersection over Union) values reported in Table 1 and used throughout our analysis are calculated based on the total number of lake pixels across the entire validation dataset. We will ensure this specific definition of MIoU is explicitly stated in the Methods section for clarity.

L270ff: To me, it was not immediately clear, why the authors chose to evaluate the performance of the models across elevation gradients. In the discussion, it turns out, that the authors associate different elevations with different environmental conditions (particularly vegetation cover and prevalence of snow). I agree, that the elevation gradient is a good proxy to model changing environmental conditions. However, I'd like a short (half-) sentence about that also in the results around L270 to avoid confusion.

**Response:** We agree that immediately clarifying the rationale for the elevation analysis will improve clarity for the reader. We will add a short explanatory phrase.

"To evaluate segmentation performance across elevation gradients, which serve as a proxy for varying environmental conditions (e.g., vegetation and snow cover) in montane regions, detection rates of Mask2Former, UNet, and DeepLab v3+ were compared (Table 3)."

Tables 4, 5 and 6: Please add the F1 score as an additional column.

**Response:** We agree and will supplement Tables 4, 5, and 6 with the **F1 score** as an additional metric, as requested.

Figure 6: While I think that the examples demonstrated here show very well the strengths of the proposed approach, for the reader it is difficult to generalize these strengths from only two samples. Consider showing 2-3 other examples for (a) and (b), respectively, as an Annex/Supplementary material to the paper to bolster your claim.

**Response:** We agree that providing more samples will better demonstrate the robustness of our approach. We will include 2-3 additional comparison examples in the Figure 6.

Section 4.2: Several things need to be addressed in this discussion:

- First, the authors need to specify, for which area the dataset comparison was conducted. Is it the STPG region again?

   **Response:** Yes, the comparison was conducted for the STPG region. We will explicitly clarify this in the revised manuscript.

- Second, when comparing their mapping results to those of existing datasets, the authors give an average size of lake area missed by the previous datasets. In terms of the relevance of very small lakes, it would also be good to know, how much of total lake area has been missed by the previous studies by including only lakes larger than a certain threshold in comparison to the newly proposed method. Similarly, it would be good to know the share of area of these very small lakes of total lake area. This way, the relevance of these small lakes would become clearer.

   **Response:** We agree that quantifying the area contribution of small lakes is crucial. We will supplement the manuscript with the total area of lakes missed by previous datasets and calculate the percentage of these small lakes' area relative to the total lake area.

- Third, while it is plausible, that the proposed method detects more lakes than the comparison datasets due to their size threshold, also the possibility of overestimation of lakes needs to be discussed. You can use the false positive rates from the results section to make an estimate.

   **Response:** While the final dataset was manually verified to minimize overestimation, we acknowledge that a few very small lakes might still present spectral confusion with similar land covers (e.g., wet soil or shadows). Due to the resolution limits and the nature of manual correction, this residual uncertainty is difficult to quantify

precisely. We will add a discussion on these potential uncertainties and the limitations of manual quality control in the revised manuscript to provide a more transparent assessment.

L419ff: As I understand it, the analysis provided here is supposed to demonstrate, how much more accurate the proposed lake classification approach is in comparison to drawing a 10 km buffer around a glaciated area and marking all lakes inside as "glacial" and all lakes outside as "non-glacial". I see several issues with this approach:

- The approach (including the selection of the buffer distance) feels arbitrary. Of course, a simple buffer, particularly one of this size, will not be able to accurately discriminate lake types. Is there previous literature that uses this approach for lake classification?

  **Response:** We clarify that the 10 km buffer distance is not arbitrary; it is a widely adopted threshold in glacial lake studies for classification (e.g., [References 1, 2, 3]). We used this established simple approach as a baseline to demonstrate the necessity and superior accuracy of our proposed classification method. In the revised manuscript, we will incorporate these justifications and references in the revised Section 4.2.

  [1] Wang X, Guo X, Yang C, Liu Q, Wei J, Zhang Y, et al. Glacial lake inventory of high-mountain Asia in 1990 and 2018 derived from Landsat images. Earth Syst Sci Data 2020;12:2169–82.

  [2] Zhang M, Chen F, Guo H, Yi L, Zeng J, Li B. Glacial lake area changes in High Mountain Asia during 1990–2020 using satellite remote sensing. Research 2022;2022:2022/9821275. https://doi.org/10.34133/2022/9821275.

  [3] Ma D, Li J, Jiang L. Efficient glacial lake mapping by leveraging deep transfer learning and a new annotated glacial lake dataset. Journal of Hydrology 2025:133072.

- Also, the selection of the region is arbitrary. Why select a single glaciated mountain range and not analyse the entire STPG region or using the validation data?

  **Response:** The selected mountain range was chosen as it is the most representative area in the STPG region to illustrate the specific challenges and limitations of the buffer-based classification. We will clarify this choice in the revised manuscript.

- Without giving any number of correctly/incorrectly classified lakes by the two approaches (similar to the tables of the results section) the performance comparison is rather meaningless.

  **Response:** We agree that numerical evidence is needed. We will supplement the analysis with a table showing the counts of correctly and incorrectly classified lakes for both methods.

However, I agree that the confusion of glacial and non-glacial lakes particularly in close proximity of glaciers needs to be addressed and evaluated! Figure 4 shows a plausible pattern of lake classifications across the region, but how robust is the proposed method specifically in regions where both types of lakes co-occur? I can imagine an accuracy assessment similar to that in Table 4 based on a subset of lakes in very close proximity to glaciers (e.g. a 1km buffer around all glaciated areas as determined by the RGI). This would be something for the results section. The discussion then needs to pick-up on these results, and, if possible, compare the performance of the proposed method (regarding lake type classification) with the comparison datasets by Zhang (2024a,b).

**Response:** We will add a specific column for lakes within 1 km proximity of glaciers to the new comparison table mentioned above. Furthermore, we will include a comparative discussion with the Zhang (2024a, b) datasets in the revised Discussion section as suggested.

Discussion section in general: Are there significant differences in computational effort between the compared DL models? If so, do the authors think the increase of accuracy is worth the additional effort?

**Response:** We agree that computational efficiency is an important consideration. In the revised manuscript, we will supplement the comparison with the specific computational time for each deep learning model. Although more complex models require slightly more processing effort, the increase in computational cost is well within an acceptable range and is significantly outweighed by the gains in mapping accuracy, which is our primary objective. We will add this discussion to the revised manuscript.

**Technical corrections**

L15-16: Add commas to sentence to enhance readability

L17-18: Suggestion: "challenges for conventional identification methods"

L26: "provided **a** more accurate lake type classification"

L80: The is a space too much here

L86: Remove full stop before citation

L97: add "the" or "a" before ViT architecture

Fig 6: Format figure caption consistently.

Fig 7: Please add scales to all of the images, and some kind of indication, where the area is located (e.g. map inset or geographic coordinates).

L394f and Table 7: Inconsistency in the citation of the Zhang (2024) papers. Please either use (2024 a/b) OR G. Zhang/T. Zhang (whichever fits best to the journal's preferred citation style).

L459: lower case o in "Overestimation"

**Response:** We appreciate these meticulous suggestions. We will address all technical corrections and editorial suggestions in the upcoming revised manuscript. Specifically, we will correct the grammatical errors, refine the terminology, and standardize the figure formatting as requested.

L164-165: The links provided refer to the data portals but not the datasets. Please provide links to the respective data catalogue entries of the platforms. If these links are to long, consider a scientific citation of the original data.

L215: Please provide direct links to the models and datasets instead of just to the platform.

**Response:** We appreciate this suggestion. In the revised manuscript, we will replace the general portal links with direct links to the specific catalogue entries of the datasets and models to ensure better accessibility.

L364: I wouldn't call the use of various spectral bands and calculated indices thereof "multisource remote sensing data", when all of the products come from a single system (Sentinel-2).

**Response:** Thank you for the correction. We agree that "multisource" is inaccurate as all data are derived from Sentinel-2. To more accurately reflect the inclusion of both original spectral bands and calculated indices, we will replace it with "multi-dimensional spectral features" throughout the revised manuscript.

---

## Author Comment (AC2)

The authors present a new method for automatic mapping and classification of high-mountain/glacial lakes applied to the Third Pole region. The manuscript is generally well composed and the method presented represents a valuable addition to the approaches applied so far. The contribution is of high relevance since knowledge on lake distribution, especially in climate-change affected mountains, is essential for hazard management and mitigation.

While the method and results are well presented and the discussion and conclusion are largely convincing, the manuscript suffers severely from a poor application of the terminology and definition of glacial and non-glacial lakes. This has little effect on the lake detection itself but huge implication on the lake classification and the results and comparison in general. With respect to the potential relevance of the produced dataset for hazard management, this issue needs to be resolved. Otherwise, despite its technological performance, the dataset will be of little use.

To conclude, I think this manuscript requires a revision with respect to the application of the right terms for the objects in focus. In addition, more attention should be paid to the introduction of the comparative database to improve clarity. These revisions require moderate effort, will not affect the geometry of the lakes dataset but surely will change the classification and the discussion. This will improve the quality of the study and ensure comparability and a wider application of the dataset in the intended way.

**Response:** We would like to express our sincere gratitude to you for the positive assessment of our technical framework and for emphasizing the importance of our work in the context of hazard management. We particularly appreciate the critical feedback regarding the terminology and definition of "glacial" and "non-glacial" lakes. We fully agree that while the deep learning model performs well technologically, the scientific utility of the resulting dataset depends on a rigorous and geomorphologically sound classification scheme. To address these concerns, we will perform a thorough revision of the terminology throughout the manuscript.

Detailed point-by-point responses to your specific comments are provided below.

Terminology: The authors need to reconsider the definition of glacial and non-glacial lakes. In the manuscript a variety of terms are applied starting with the term periglacial lakes in the title and introduction (and not more afterwards) than glacier lakes, montane lakes and non-glacier lakes. The authors mention to follow the classification by Yao et al. (2018) but a detailed definition of the terminology is absolutely required. This will influence the results and interpretation. For additional clarification I suggest fundamental review papers on the terminology for example by Carrivick and Tweed (2013) [DOI: 10.1016/j.quascirev.2013.07.028]

Furthermore, the title is confusing. Despite the use of the term "periglacial lake", I also don't know what "global attention" is signifying in this context. Please reconsider a more appropriate title.

L39ff - You should provide a better definition of non-glacial lakes. The reference to "thermodynamic processes" is not enough from a geomorphological perspective since this is a too broad term from physics. The term periglacial lakes is not commonly used, since the formation is not linked to periglacial processes (involving ground ice and freeze-thaw). Using periglacial lakes with respect to the location of the lake should be avoided due to the misleading connotation of the term periglacial here.

**Response:** We sincerely appreciate the reviewer's guidance on terminology. We have carefully studied the suggested literature by Carrivick and Tweed (2013) and acknowledge that our initial use of "periglacial lakes" as a single category was not sufficiently rigorous.

In the revised manuscript, we will clarify that our study focuses on "lakes in periglacial environments." Within this environment, we will implement a binary classification:

Glacial Lakes: We will adopt the definition by Yao et al. (2018): "natural water bodies mainly supplied by modern glacial meltwater or formed in depressions of glacial moraines." The primary reason for selecting this definition is that our manual interpretation and data labeling were strictly conducted according to these established criteria. To ensure absolute consistency between our methodology and our scientific definitions, we believe this approach is the most appropriate and robust.

Non-glacial Lakes: All other water bodies located within the periglacial study area that do not meet the above criteria will be categorized as non-glacial lakes.

Furthermore, we agree that "Global Attention" could be misinterpreted as "global interest" from the public. To clarify that our study focuses on the comprehensive spatial information (long-range dependencies) captured by the Transformer model, we will replace this term with "Global Context" throughout the text.

In addition, acknowledging the feedback from another reviewer that "Empowers" might be overly strong, we have opted for the more precise term "Enhancing." Consequently, we have synthesized all suggestions and will revise the title to:

"Enhancing Lake Identification in Periglacial Environments by Leveraging the Global Context of Transformers"

To illustrate this, one must investigate chapter 3.3: In the STPG region most of the non-glacial lakes identified and depicted in Fig. 4 are indeed glacial lakes, according to most

classification schemes, because they have been formed by glacial erosion. Many are found in cirques that have been sculpted by glaciers (e.g. in the area around 29°,11.441' N/95°33,340'E). The only difference is that they are located in catchments without current glaciers, thus they have been formed by glacier action in the past. Your terminology should therefore not only include a geomorphological and topographic definition, but also a temporal one (see for example Buckel et al. (2018)).   Non-glacial, from my perspective would be restricted to lakes formed by landslides/debris flow dams or of volcanic origin. Lakes purely formed by excessive precipitation are vary rare in mountainous regions from my perspective.

My suggestion would be to either add a temporal aspect to your definition (Holocene, historic glacial lake) or to only to focus on ice-contact or near-glacier lakes (which would involve a distance-based definition).

This terminological uncertainty should be resolved and then considered in the discussion of the distance-based method. Your comment may of course be valid for some applications esp. natural hazards assessment (e.g. GLOF), but some of the argumentation is lost when the terminology is better defined and applied. In this respect authors need to consider that the distance-based method is justified here, assuring that there is a glacier upslope of the lake.

**Response:** We appreciate the reviewer's insightful comment regarding the geomorphological origin of these lakes. We acknowledge that many lakes in cirques were historically sculpted by glacial action. However, as our study follows the classification framework of Yao et al. (2018), our distinction between "glacial" and "non-glacial" is primarily based on modern hydrological processes and contemporary glacial influence rather than long-term geomorphological evolution.

To address the reviewer's concern, we will explicitly state that our "glacial lakes" are those influenced by modern glaciers (current meltwater or moraine proximity). Lakes in relict glacial landforms (like old cirques) without current glacier coverage are classified as "non-glacial" in this study to maintain consistency with our interpretation criteria. This definition is particularly valid for practical applications like GLOF hazard assessment, which focuses on lakes with active glacier-lake interactions.

We clarify that the 10 km buffer distance is not arbitrary; it is a widely adopted threshold in glacial lake studies for classification (e.g., [References 1, 2, 3]). We used this established simple approach as a baseline to demonstrate the necessity and superior accuracy of our proposed classification method. In the revised manuscript, we will incorporate these justifications and references in the revised Section 4.2.

[1] Wang X, Guo X, Yang C, Liu Q, Wei J, Zhang Y, et al. Glacial lake inventory of high-mountain Asia in 1990 and 2018 derived from Landsat images. Earth Syst Sci Data 2020;12:2169–82.

[2] Zhang M, Chen F, Guo H, Yi L, Zeng J, Li B. Glacial lake area changes in High Mountain Asia during 1990–2020 using satellite remote sensing. Research 2022;2022:2022/9821275. https://doi.org/10.34133/2022/9821275.

[3] Ma D, Li J, Jiang L. Efficient glacial lake mapping by leveraging deep transfer learning and a new annotated glacial lake dataset. Journal of Hydrology 2025:133072.

Some minor comments:

L36– exchange the term "montane" with "alpine/high-alpine" – montane refers a biogeographic altitudinal zone usually at intermediate altitudes. (throughout the manuscript!!)

**Response:** Thank you for the correction. We agree that "alpine/high-alpine" is more accurate for the high-altitude context of our study. We will replace "montane" with "alpine" throughout the manuscript and in the revised title.

L58ff – same issue as above…

**Response:** We agree with the reviewer's concern regarding the terminology and the vague definition of "non-glacial lakes." In the revised manuscript, we will remove the term "periglacial lakes".

Following the classification criteria of Yao et al. (2018), we will provide a clear, binary definition: Glacial lakes are explicitly defined as "natural water mainly supplied by modern glacial meltwater or formed in glacier moraine's depression." Consequently, non-glacial lakes are defined as any lakes that do not meet these specific criteria (i.e., lacking modern glacial influence). These clarifications will be implemented throughout the paper to ensure geomorphological and terminological rigor.

L251ff – You compare the result to other approaches (CNN, UNet, DeepLapv3+), but you don't mention that you applied these methods as well. How was this comparison done? Did you use existing data from other studies? This need to be mentioned in the methods section (e.g. 2.5) and reference in Table 1.

L282 – Ch 3.2 – Similar to the comment above – You compare your classification results with two other CNN approaches (EfficientNet, ResNet). How was this done? Again no mentioning in the methods before.

**Response:** We clarify that the results of the other approaches (CNN, UNet, and DeepLabv3+) were not obtained from previous studies; instead, we implemented and trained these models

ourselves using the same dataset and basic parameters to ensure a fair and direct comparison. Following your suggestion, we will add a detailed description in the Methods section explaining the implementation, training environment, and parameter settings for these comparative models.

L269 – Table 2 (and same for table 3): The tables hold the category "all". What does this mean? Are these the mapped lakes? I suggest renaming this class for better clarity.

**Response:** We agree that the category "All" is ambiguous. In the revised manuscript, we will rename this class to "Ground Truth" in both Table 2 and Table 3 to clearly indicate that these represent the reference data used for verification.

L291 – Add explanation for TP, FP, TN, FN in the table caption.

**Response:** We will address these as suggested in the revised manuscript.

L329 – Exchange "The proposed framework" with a more precise description excluding the CNN/alternative methods. Like: ViT-based methods…

**Response:** We will replace "The proposed framework" with a more precise description, such as "the ViT-based identification method," as suggested.

L395ff – Chap 4.2 – please add the a, b to the Zhang references throughout the chapter to better differentiate between the publications.

**Response:** We will address these as suggested in the revised manuscript.